# An Overview of Management Status and Recycling Strategies for Plastic Packaging Waste in China

Chaojie Yu [1], Diyi Jin [1], Xichao Hu [2], Wenzhi He [1] and Guangming Li [1,*]

[1] College of Environmental Science and Engineering, Tongji University, Shanghai 200092, China; 2030524@tongji.edu.cn (C.Y.); jdy_2509568374@163.com (D.J.); hithwz@163.com (W.H.)
[2] Shanghai Tianqiang Environmental Technology Co., Ltd., Shanghai 201415, China; 15801871619@139.com
[*] Correspondence: ligm@tongji.edu.cn

**Abstract:** Given their exceptional performance, plastic packaging products are widely used in daily life, and the dramatic expansion in plastic packaging waste (PPW) has exacerbated environmental problems. Many countries have enacted laws and developed recycling technologies to manage plastic packaging waste in consideration of the nature of PPW as both garbage and a resource. As the world's largest producer and consumer of plastics, China has also taken measures to address this issue. This paper presents the latest management regulations and recycling strategies for PPW in China. Based on an analysis of the current management status of PPW and recycling technologies and their carbon emission impacts, some management suggestions and a comprehensive full-chain recycling process were put forward. We supposed that management challenges that need to be overcome in the future can be solved through the improvement of green designs for plastic packaging, manufacturing technology updates, consumption concept changes, and the high-value utilization of PPW. This paper aims to provide valuable references for government decisions on PPW management and, furthermore, to set up an economically sensible and industrially feasible PPW solution and boost the development of PPW recycling.

**Keywords:** plastic packaging waste; regulations; carbon reduction; full-chain recycling process; management suggestions; China





## 1. Introduction

Plastic is an innovative material known for high ductility, durability, stability, lightweight, and portable features [1]. It finds extensive applications in packaging, automobiles, electrical appliances, furniture, and many other industries [2,3]. Since the 1950s, the commercial production of plastics has experienced extraordinary growth [4], reaching 367 million tons/year in 2020 [5]. After decades of development, China's plastics industry has evolved into a comprehensive, independent manufacturing system that plays a significant role in the national economy. As shown in Figure 1, China's plastics production has consistently remained high and experienced steady growth in recent years [6].

As one of the most important application scenarios for plastics, the production and utilization of plastic packaging have significantly improved people's living standards and promoted economic and social development. However, plastic packaging products have been developed within a "linear economy" approach in various aspects of human life, neglecting their secondary use or carbon emission impacts [7]. Consequently, escalating amounts of plastic packaging waste (PPW) and its additives are being discarded into the environment, causing increasingly severe damage to the ecosystem and living organisms [8,9]. Plastic products are made from non-renewable resources such as crude oil; recycling PPW can increase the total utilization rate of natural resources, raise the income level of inhabitants, and reduce waste plastic pollution in the ecological environment [10,11]. This also constitutes a crucial aspect of the sustainable growth of plastics-related enterprises.

Furthermore, it is essential for the long-term development of plastics-related sectors. On the whole, recycling PPW can not only mitigate environmental pollution and achieve sustainable development but also yield substantial economic advantages.

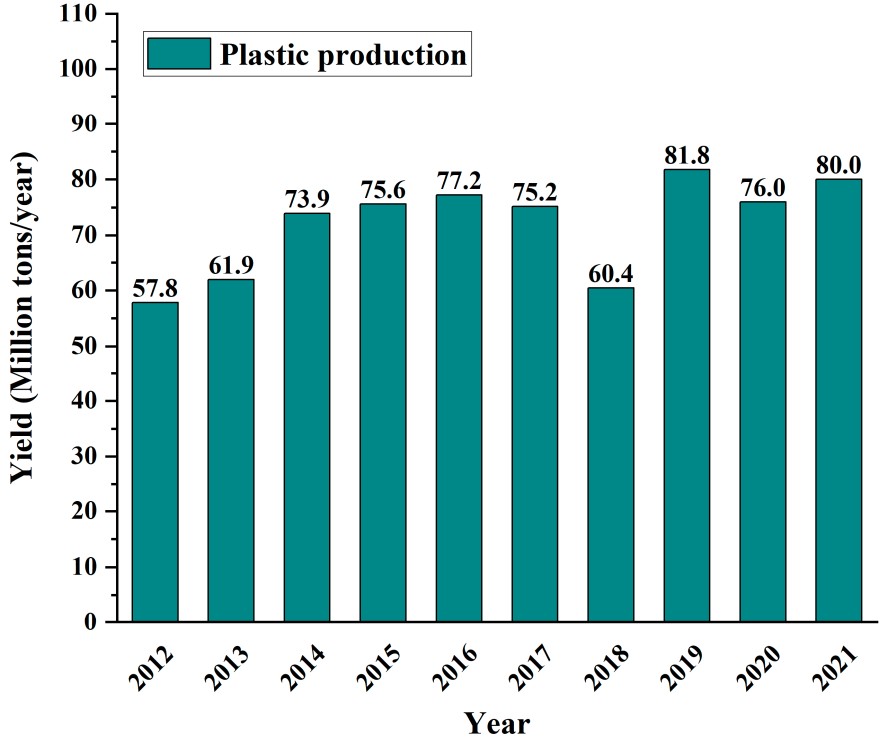

**Figure 1.** The yield of plastic products in China in the past decade.

Considering the enormous quantity, environmental risk, and resource properties mentioned above, many countries have issued a series of laws and regulations to manage PPW and achieved remarkable results [12,13]. To reduce PPW and enhance recycling efficiency, the EU issued the EU Plastics Strategy [14] in January 2018, involving an investment of EUR 350 million to modernize plastics production and recycling. The strategy aims to make all plastic on the EU market reusable or recyclable by 2030 and raise the recycling rate to 55%. In November 2021, the US introduced its first National Recycling Strategy (NRS) [15], focusing on strengthening and advancing municipal solid waste (MSW) recycling systems, with a strategic goal of increasing the recycling rate of PPW to 50% by 2030. The NRS outlines five action plans, including enhancing markets for recycled plastic products, expanding waste plastic recycling and reclamation infrastructure, reducing impurities in waste plastics, and strengthening PPW recycling and reclamation infrastructure. The five action programs are improving markets for recycling plastic products; boosting recycling and recycling infrastructure; lowering contaminants in plastics; strengthening policy and standardizing analyses of plastics recycling; and collecting data. In January 2001, the Japanese government enacted the Basic Law for the Promotion of a Recycling Society [16], establishing a basic framework for recycling-oriented social regulations and exploring a circular economy for a recycling society based on the principles of "reducing, reusing, and recycling" (the 3Rs). The National Plastic Resource Recycling Strategy [17] was formulated in May 2019. At the G20 Summit held in Osaka in June of the same year, managing marine plastic debris and microplastics was set as a fundamental problem. Australia launched the National Plastics Plan in 2021 [18], which outlines national plastic pollution control actions and sets out the timetable for control tasks from 2019 to 2030, promoting plastic pollution control actions in five aspects: source prevention, end-of-life recycling, public participation, marine plastic control, and scientific and technological innovation. The goal is to ensure that 70% of plastic packaging is recyclable, reusable, or compostable by 2025.

Being the largest producer and consumer of plastics globally, China has implemented necessary measures to deal with PPW management [19–21]. Recently, a series of policies were introduced in China to promote plastic waste recycling. In 2020, the National Development and Reform Commission and the Ministry of Ecology and Environment of China further issued the Opinions on Further Strengthening the Prevention and Control of Plastic Pollution [22]. These guidelines advocate for a gradual reduction in non-essential single-use plastic products, such as disposable plastic bags, lunch boxes, and toiletries. They also emphasize the necessity of enhancing production, consumption, recycling, and disposal mechanisms for plastic products. Therefore, assessing the future management of China's waste plastic becomes crucial for the effective implementation of these policies.

Despite the high processing rate (processing volume/waste plastic volume) of PPW in China, reaching up to 98%, over half of PPW is still disposed of in simple ways such as landfills and incineration [23]. These practices result in secondary pollution to the environment and enormous carbon emissions. In recent years, the recycling rate (recycling volume/PPW volume) has remained at approximately 27% [24]. Most PPW recycling enterprises in China are "small workshop" enterprises. They rely on manual sorting methods and generally poor recycling and sorting levels and profitability, making it challenging to ensure a stable supply of PPW in terms of quantity and composition. Given the impact of the COVID-19 pandemic, the recycling of PPW has ground to a halt, causing a dilemma where some PPW recycling enterprises lack available materials. The recycling utilization rate of waste plastics dropped for the first time in 2020, from 30% in 2019 to 27% in 2020 [25]. China has plenty of room to improve its PPW management to meet the increasing demand for high-quality plastic products and comply with the increasing environmental protection policies.

This paper presents a current overview of the framework for managing PPW in China, as well as an exploration of PPW recycling technologies and challenges related to carbon emissions. Based on the current management status and recycling technologies for PPW in China, some management suggestions and an environmentally friendly and industrially feasible full-chain recycling process for PPW were put forward.

## 2. Management Status of PPW in China

Currently, China is the largest producer and consumer of plastics in the world [26]. However, China began managing PPW later than developed nations, resulting in an inefficient PPW management system [27]. China has issued specific and restrictive regulations to guide and standardize PPW recycling over the past several decades [28,29]. The following table (Table 1) summarizes the most important Chinese regulations regarding PPW recycling. During the infancy of the plastics industry, the vast majority of pertinent policies targeted the management of single-use foamed plastics [30]. As science and technology advanced and the importance of the concept of the circular economy grew, the plastics industry was gradually proposed as a significant contributor to the circular economy. Efforts have been directed toward the efficient recycling of PPW and comprehensive waste management policies [31–33].

China's comprehensive utilization of the waste plastic resources industry has led to the establishment of a regulatory framework that combines government oversight with self-regulatory management by industry associations [34,35]. Key industry authorities in this regulatory system include the State Administration of Market Supervision and Administration, the National Development and Reform Commission, the Ministry of Commerce, the Ministry of Industry and Information Technology, the Ministry of Ecology and Environment, and the Ministry of Housing and Urban-Rural Development. These agencies are responsible for developing and coordinating the implementation of industrial policies and industry planning, as well as drafting relevant laws, regulations, and rules.

Three prominent industry associations, namely, the China Plastics Processing Association, the China Synthetic Resin Association, and the China Material Recycling Association, play vital roles in assisting the formulation of industry policies and standards. They also

contribute significantly to the collection of data pertaining to the production, utilization, and disposal stages of waste plastics [36].

**Table 1.** Legislations related to PPW in China.

| Laws and Regulations | Year | Major Regulations on Plastic Recycling |
|---|---|---|
| Urgent notice on the immediate cessation of the production of disposable foamed plastic tableware | 2001 | Stop the production of disposable foamed plastic tableware immediately and look for suitable alternatives. |
| A Notice on Restricting the Use of Plastic Shopping Bags for Production and Sale | 2008 | The production, sale, and use of plastic shopping bags with thicknesses less than 0.025 mm are prohibited; shopping bags are purchased for use. |
| Soil Pollution Control Action Plan | 2016 | Strengthening the recycling and utilization of waste agricultural film. |
| The Implementation Plan for Prohibiting the Entry of Foreign Garbage and Promoting the Reform of the Solid Waste Import Management System | 2017 | PPW imported from domestic sources will be banned at the end of 2017; PPW from industrial sources will be banned in 2019. |
| Pilot Work Plan for the Construction of "Waste-Free Cities" | 2018 | Establish an index system for "waste-free cities"; coordinate the management of solid waste in economic and social development. |
| New Law of the People's Republic of China on the Prevention and Control of Environmental Pollution by Solid Waste | 2020 | Encourage and guide reductions in use; actively recycle plastic bags and other disposable plastic products; and promote the application of recyclable, easily recyclable, and degradable alternative products. |
| Opinions on Further Strengthening the Control of Plastic Pollution | 2020 | Promote alternative products; standardize the recycling of waste plastic packaging; improve the management system of the production, circulation, use, recycling, and disposal of plastic products. |
| "14th Five-year Plan" Plastic Pollution Control Action Plan | 2021 | By 2025, make full-chain plastic product production, circulation, consumption, recycling, and end disposal more effective. |
| Notice on the Issuance of Action Plans for the Treatment of New Pollutants | 2022 | Formulate "one product, one policy" control measures for microplastic pollution. |

In response to the rapid growth of the plastics industry, China's management policy system for PPW has gradually become more precise, and relevant regulations continue to be implemented [37]. In recent years, the policy and regulatory framework of PPW management in China has gradually taken shape. In 2017, the Chinese government introduced the "Household Waste Separation System Implementation Plan." Subsequently, numerous cities initiated their implementation plans in April 2018 and further refined them at the local level. The creation and implementation of the Implementation Plan for the Domestic Waste Separation System significantly impacted the management of solid waste, including plastic waste, in China. Following multiple revisions and consultations, the Chinese government reissued the "New Law of the People's Republic of China on the Prevention and Control of Environmental Pollution by Solid Waste" (henceforth referred to as the "New Solid Waste Law") at the conclusion of April 2020. This law has become a crucial guiding principle for managing plastic waste in China. In Figure 2, the thick black line represents the relationship between the superior and subordinate laws, while the thin black line represents the guiding significance. For instance, the new Solid Waste Law is superior to numerous new plastic management regulations. Concurrently, the domestic waste separation system's implementation plan has vital guiding significance for the new Solid Waste Law.

Since China proposed its carbon peaking and carbon neutrality goals in September 2020, the emphasis on waste recycling has increased, and the management of PPW has become more precise [38]. Regarding plastic restrictions, the National Development and Reform Commission and the Ministry of Ecology and Environment issued the Opinions on Further Strengthening Plastic Pollution Control in January 2020. This policy specifies the types of plastic packaging products that will be banned and restricted, as well as a withdrawal schedule. In 2021, the National Development and Reform Commission issued both the "Guidance on Expediting the Establishment of a Green, Low-Carbon, Circular

Economic System" and the "14th Five-Year Plan for Circular Economy Development"; these directives set forth requirements for the management and recycling of plastic packaging waste that exceed international standards [39]. These directives underscore the imperative to accelerate the development of a waste materials recycling system grounded in the principles of reduce, reuse, and recycle (the 3Rs).

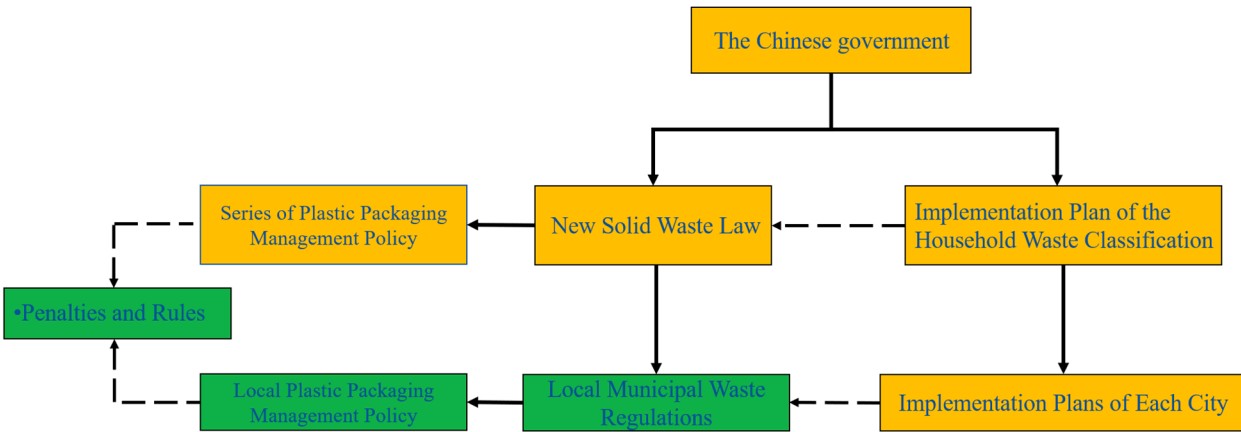

**Figure 2.** Policy framework related to PPW management in China.

The implementation of pertinent policies has yielded certain results; China has gradually established a comprehensive plastic waste recycling and utilization system that spans across society. According to estimates from relevant industry associations, in 2021, China's recycling volume of plastic waste materials reached approximately 19 million tons, with a material recycling rate of 31%; this rate is 1.74 times higher than the global average, achieving 100% domestic material recycling utilization [40]. In comparison, during the same period, the domestic material recycling rates in the United States, the European Union, and Japan were only 5.31%, 17.18%, and 12.50%, respectively [41]. In China, the volume of plastic packaging exports far surpasses that of other types of packaging. Specifically, plastic packaging accounts for 65.91% of the total export value, whereas, in imports, it constitutes 90.13% of the total import value [42].

In the realm of standardization, China has been progressively enhancing its efforts. Since May 2021, a series of national standards, including "Plastic Recycled Plastics Part 1: General Rules" and 'Plastic Recycled Plastics Part 2: Polyethylene (PE) Materials', have been introduced to regulate and govern the industry. These standards signify substantial institutional reforms aimed at promoting sustainable, environmentally friendly development, thus preserving ecological balance and safeguarding human well-being [43]. They can reduce the amount of domestic PPW and standardize the market for recycling domestic waste plastics. Because of these policies, China's waste recycling industry can enter a new development phase.

The country is conducting extensive research on PPW processing in terms of scientific and technical support. The key project regarding "solid waste recycling" of the National Key Research and Development Plan deployed research on "green recycling and high-quality utilization technology of waste and miscellaneous plastic packaging" in 2019, followed by research and development tasks related to a new microplastics compound pollution transmission mechanism and blocking technology in 2020.

According to the Department of Circulation Industry Development, the domestic waste plastic recycling volume in 2019 was $1.89 \times 10^7$ t, with a recycling rate of nearly 30% and a total recycling value of more than CNY 100 billion [44]. China's waste plastic recycling field contains more than 10,000 enterprises, but the vast majority of them are small and micro-enterprises [45]. The relative scarcity of large-scale enterprises has contributed to an insufficient recycling network for waste plastics, resulting in suboptimal recycling rates and causing market fluctuations because of unstable material supply. Nevertheless,

the broader perspective reveals that China's PPW recycling industry has succeeded in establishing relatively centralized recycling trading markets and processing distribution centers. The industry as a whole is progressing toward greater market orientation and scale [46].

As demonstrated in the previous section, China has made significant changes and advances in PPW recycling policy management, demonstration bases, and investments in science and technology. Under the influence of carbon peaking and carbon neutrality goals, China's PPW recycling industry is expected to enter a new phase of development and become one of the largest PPW recycling markets in the world [47].

## 3. Recycling Technologies and Carbon Emission Impacts of PPW

Plastic packaging is widely used in China, with packaging plastics accounting for nearly 50% of the total production [20]. The industrial chain of plastic packaging can be divided into three parts: the upstream is the supplier of raw materials for plastic packaging, providing synthetic resins for plastic packaging raw materials; the middle stream is mostly the sale of plastic products; and the downstream is the application and recycling industry for plastic packaging. The recycling techniques for PPW are generally categorized into mechanical recycling, chemical recycling, and energy recycling [11,48]. Furthermore, based on variations in recycling grades, these techniques can be further subdivided into four distinct recycling levels [49], shown in Figure 3. Mechanical recycling primarily involves the physical treatment of PPW (Section 3.1), and the vast majority of PPW in China is treated in this way. Depending on the cleanliness of the PPW, mechanical recycling can be categorized into the primary level and the secondary level. Chemical recycling corresponds to the tertiary level of recycling, which recycles the components in PPW and transforms them into chemicals or raw chemical materials such as oil (Section 3.2). Energy recycling is the quaternary level achieved through the combustion of PPW (Section 3.3). Increasing demand for sustainable development and the adoption of green technologies have also heightened the significance of assessing carbon emissions generated during the recycling process of PPW (Section 3.4).

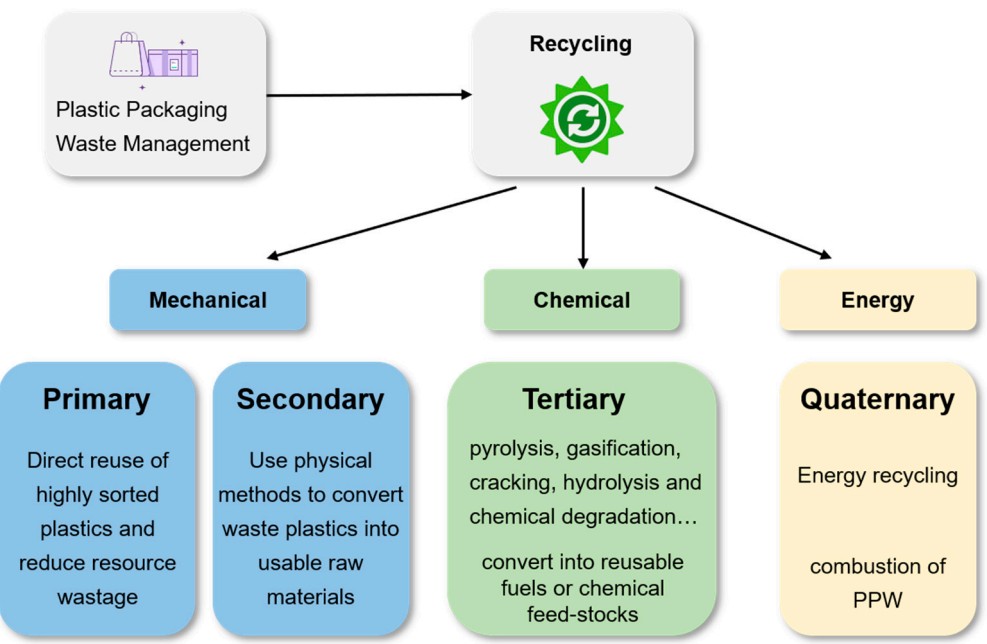

**Figure 3.** Overview of PPW recycling techniques.

### 3.1. Mechanical Recycling

Mechanical recycling plays a crucial role in the global waste plastic recycling industry [50]. The mechanical recycling of PPW refers to the physical reprocessing and

reforming of PPW into new marketable products [51]. This method finds extensive application in China's PPW recycling sector, with well-developed technologies and established markets [11,52].

Depending on the process employed, mechanical recycling methods can be categorized into two main levels: simple recycling at the primary level and modified recycling at the secondary level [53]. The simple recycling method involves melting and molding PPW immediately after sorting and cleaning it. The simple recycling approach involves the utilization of a simplified production process and equipment to create a new plastic product with an identical chemical composition to that of the original waste plastic. However, this process is associated with diminished physical properties, such as mechanical parameters [54]. Mechanical treatment steps include cutting, floating, washing, drying, extrusion, and collision to process PPW [50,55]. Processes such as cross-linking, grafting, and chlorination can add chemicals to enhance waste plastic's physical and chemical properties [56,57]. Modified recycling can increase the value of PPW by enhancing one or more of its properties. Nonetheless, this method necessitates a more complicated production process, specialized equipment, and substantial capital investment [58].

Mechanical recycling is widely used around the globe to treat PPW because of its low cost, ease of operation, and replicability [59]. Nevertheless, there exist significant impediments to the progress of this technology, including issues such as the suboptimal performance of recycled materials, restricted application scopes, limited recyclability beyond one or two cycles, and dependence on recycling and sorting capacities. Presently, advanced research is centered on the development of high-value mechanical regeneration products with enhanced economic potential [60,61]. For example, Xu [62] studied the properties of plastics obtained by mixing and modifying recycling polypropylene plastic (RPP) with different co-binders, such as polypropylene new material (VPP) and ethylene-octene copolymer (POE). The progressive advancement of mechanical recycling is gradually phasing out outdated production capacities [63].

*3.2. Chemical Recycling*

PPW contains a significant amount of energy and polymer molecules, which can be used as a source of raw materials for producing chemicals and fuels [64]. Chemical recycling (considered tertiary recycling) converts used plastics into reusable fuels (gasoline, diesel) or chemical feedstocks (ethylene, propylene) using cracking technology [65]. Chemical recycling is in line with the principle of sustainability and can result in products with relatively high added value [66–68].

In the field of the chemical recycling of PPW, pyrolysis is considered one of the most researched technologies, where high temperature and pressure conditions are used to produce the corresponding hydrocarbons. Especially with catalysts, the temperature and pressure required for the reaction can be reduced, and the product composition can also be regulated to produce high-quality products on demand. Corresponding catalyst development is in progress to address the high temperatures required, long reaction times, low yields, and generation of hazardous emissions. An efficient method for pyrolysis and recycling PPW from hybrid vehicles has been reported, resulting in pyrolysis oil with a heat content of 43,806 J/g. Its physical and chemical properties were comparable to those of motor gasoline [69]. There are many other chemical recycling methods, including gasification, hydrolysis, and chemical degradation [70,71]; mechanistic studies of these methods are also being carried out [72].

For PPW that cannot be processed with mechanical recycling, the use of chemical recycling to convert it into clean oils or chemical raw materials represents a viable approach to the recycling of PPW. Chemical recycling offers distinct advantages when handling PPW from diverse sources [73], as it necessitates a minimal raw material pre-treatment and reduces the labor and material inputs required for initial sorting and cleaning processes [74]. In recent years, chemical recycling technology has advanced rapidly [75–77]. However, given the complexity of chemical recycling equipment and its high energy consumption, it



has been deemed difficult to promote this application economically [78]. Developed nations have sought cost reduction through scale, with international chemical industry leaders taking the initiative in industrialization layouts [79]. the development of chemical recycling in China is relatively lagging behind, with only Sinopec and a few private enterprises having carried out tonnage scale-up or pilot studies on the chemical recycling of PPW.

*3.3. Energy Recycling*

Energy recycling, combustion into recycled thermal energy, primarily applies to heavily polluted PPW that conventional physical and chemical methods cannot recycle [80]. Waste incineration generates high-temperature gas for power generation. In contrast, incineration produces hydrogen chloride, dioxins, polycyclic aromatic hydrocarbons, and other toxic gases, resulting in secondary air pollution [48]. Research indicates that developing advanced green high-temperature incineration equipment should be accelerated to achieve safe and clean incineration [81]. In 2021, 27.6 million tons of plastic waste in China were incinerated along with household garbage [82], with the full potential of energy recycling efficiency yet to be fully realized.

Moreover, during the COVID-19 pandemic, the sensible energy recycling of PPW is of the utmost importance. The demand for plastics for pandemic protection can be summed up as an increase in plastic consumption attributable to PPE (PPE—personal protective equipment) [83]. The majority of plastics used in this manner are non-biodegradable plastics. These materials are primary components of protective clothing, garbage bags, plastic packaging, masks, goggles, gloves, and hand sanitizer bottles. If the global population adheres to a single disposable mask daily, there would be a monthly demand for 129 billion masks and 65 billion gloves [84]. According to China's Ministry of Ecology and Environment, Wuhan generated approximately 240 tons of medical waste per day during the pandemic, with a high percentage of plastic, compared with 40 tons per day before [85]. According to an analysis conducted by the World Economic Forum, the United States could generate an entire year's worth of medical waste (mainly plastic packaging waste) in two months because of the impact of COVID-19 [86]. According to some studies, plastics can be carriers of the COVID-19 virus, which can even survive for up to 72 h [87]. Therefore, energy recycling has become a secure and efficient method of handling medical PPW.

Energy recycling for heat and power generation is an effective way to achieve waste reduction for urban solid waste, including PPW. This approach has been widely adopted by countries worldwide [88]. By eliminating the need for the complex pre-sorting processes of old plastics, direct incineration can handle large quantities of plastic waste and municipal solid waste efficiently. The utilization of heat energy from incineration aligns well with the current challenges in waste classification faced by China, where the effectiveness of waste sorting is limited.

The efficiency of energy recycling is influenced by the recycling equipment used. According to the consulting firm Eunomia, the efficiency of generating heat energy by incinerating plastics in incinerators used for power generation is 25%, whereas new advanced gas-fired power plants operate at 55% efficiency [89]. Through methods such as equipment upgrades, improving the efficiency of PPW energy recycling can greatly enhance its competitiveness.

In China, energy recycling has become the second major method for waste disposal [90]. By the end of 2021, approximately 743 waste-to-energy plants were operational in the country, with a total processing capacity of around 820,000 tons per day [91]. Several policy measures have been implemented in China, such as mandating the use of waste heat recovery in municipal solid waste incineration and providing incentives for waste-to-energy electricity generation, all aimed at promoting the energy recycling of waste, including PPW. However, energy recycling still faces various challenges in China, including chlorine element pollution, high investment and operational costs, and low adaptability to imported foreign technologies and equipment [92].

### 3.4. Carbon Emission Impact Assessment

At present, the recycling of PPW is recognized as a sustainable practice because of its contribution to reducing the necessity for new plastic production, thereby mitigating carbon emissions [50,93]. The carbon emissions associated with the production stage of plastic packaging, which is a significant petrochemical product, have been extensively investigated [94]. In contrast, the carbon emissions of the PPW recycling stage only gained attention at a later time but have since become a focal point of research.

Methods of evaluating the carbon emission impacts of PPW recycling have continued to advance, with studies revealing the necessity of considering not only the direct emissions from the recycling process but also other factors throughout the entire recycling life cycle [95]. This holistic view encompasses post-recycling plastic reprocessing, transportation, storage, and final product manufacturing. Consequently, a comprehensive assessment of carbon emissions in PPW recycling requires the meticulous consideration of multiple phases to attain a comprehensive understanding of its true environmental performance.

Carbon emission accounting encompasses various methods, including life cycle assessment (LCA), input–output analysis (IOA), energy fossil fuel emission calculation (EFEC), and the Kaya carbon emission constant equation (KCECE), among others [96]. Assessing a product's carbon footprint through the LCA approach enables the comprehensive evaluation of greenhouse gas (GHG) emissions associated with a specific product throughout its life cycle stages [97–99]. Typically, for carbon footprint calculations, the global warming potential (GWP) values outlined in the Guidelines for National Greenhouse Gas Inventories by the United Nations Intergovernmental Panel on Climate Change (IPCC) are employed. These values are expressed in terms of $CO_2$ equivalents, with the unit of measurement as kg $CO_2$-eq. In recent years, research efforts have increasingly focused on carbon emission accounting for the mechanical and chemical recycling processes of PPW [67,99–102]. Energy recycling through incineration serves as a baseline value for assessing the carbon emission impacts arising from different mechanical and chemical recycling processes.

The evaluation of carbon emissions from the mechanical recycling process of PPW is influenced by various factors, including the target population, the scope of the study, and the comprehensiveness of the data inventory. An investigation was carried out to analyze the key factors influencing carbon emissions for different categories of recycled PPW products, encompassing roof coverings, plastic pellets, pallets, and films. The findings indicate that the cleaning and granulation processes conducted by recyclers have the most substantial influence on overall greenhouse gas (GHG) emissions [103]. The total carbon footprint of LDPE under the recycling scenario in Korea was calculated to be $6.82 \times 10^3$ kg $CO_2$-eq. with the LCA method using 400,000 pieces of LDPE plastic sheet with dimensions of $300 \times 250 \times 0.06$ mm as a functional unit [104]. Notably, the carbon footprint of LDPE recycling is an order of magnitude lower than that of landfilling and incineration, underscoring its commendable environmental friendliness. Another study employed real-world data from an urban setting to develop a comprehensive life cycle inventory and assessment model. The results of the model indicated that, in a representative urban environment with limited waste management options, reusable polypropylene nonwoven bags (PNBs) exhibit a lower overall negative environmental performance compared with the use of high-density polyethylene (HDPE) bags, biodegradable plastic bags, and kraft paper bags [105].

The evaluation of carbon emissions from PPW's chemical recycling process is relevant to factors such as recycling technology and types of recycled products. By analyzing 75 waste treatment options, it was found that chemical recycling into monomers or value-added products has the potential to reduce global warming impacts [79]. Recycling low-value PPW could reduce greenhouse gas emissions from +153.8 to −112.7 $CO_2$-eq. per ton compared with incineration [106]. Other studies have found that increased olefin production through carbon capture and utilization and by-product conversion can further reduce GHG emissions [73,107]. There is limited research specifically addressing the carbon emissions impact of plastic packaging waste (PPW) in China. Some scholars have

conducted a carbon footprint assessment of a particular company's plastic composite packaging materials. Data collection was performed on packaging materials with dimensions of 245 mm × 322 mm and a volume of 1 L. The calculation revealed that 1 L of plastic composite packaging material generates a carbon footprint of 69.9 g [108]. It was observed that the raw material production process contributes significantly to the carbon footprint, and as the individual packaging volume increases, the carbon footprint also increases. Conversely, the carbon footprint decreases with an increase in packaging volume per unit.

Given the carbon emission impacts of PPW, there is a growing tendency in the industry to replace plastic packaging with paper-based alternatives (such as cardboard) [109] to reduce pollution and carbon emissions. However, when considering the density ratio between paper and plastic, paper, being heavier, results in a higher carbon footprint for the same product packaging compared with plastic [110]. According to a 2022 report by McKinsey, paper grocery bags have double the production emissions of HDPE bags because of their higher raw material usage and transportation emissions [111]. There have been studies comparing the carbon emission impacts of various alternative materials and plastic packaging [112,113]. Overall, a particularly suitable substitute material has not been identified yet.

In recent years, China has implemented stringent policies regarding the control of imported recycled plastics, placing a strong emphasis on the utilization of high-quality recycled plastic packaging materials. Additionally, a unique and specific evaluation system is being developed to analyze the carbon emissions of PPW recycling. It is anticipated that, in the future, China will engage in the recycling and better use of PPW while adhering to standards of quality and sustainability, ultimately working toward the objective of reducing carbon emissions.

## 4. Recycling Strategies of PPW in China

### 4.1. Green Design of Plastic Packaging

The design of many plastic packaging products is currently diverse and complex in terms of materials, colors, labels, and even shapes in order to meet consumer demand and ensure product novelty [114,115]. These factors significantly influence the efficiency and quality of post-waste recycling and the improvement of the waste plastic recycling rate. Likewise, PPW recycling typically requires de-mixing and sorting, and the use of materials and labels for plastic products significantly impacts this procedure [116–118]. For instance, the metal materials in plastic products should be sorted or separated, and the trademarks of different materials on the surface of plastic products should be peeled off; the binder used to affix the trademarks affects the ease of peeling, and the use of harmful substances affects the environmental friendliness of the product [70]. Hence, it is evident that the selection of plastic products and their other accompanying constituent materials impact their recycling.

China has long advocated for eco-friendly design principles, which have found applications in various industries. However, marketing plastic products has proven to be challenging given their intricate nature [119]. To address this issue effectively, a strategy that commences with a waste plastic recycling system and a regeneration phase, while also considering the ease of recycling and regeneration during the design of plastic products, holds the potential to significantly increase recycling rates and reduce the generation of PPW [120].

The Green Recycled Plastics Supply Chain Joint Working Group (GRPG) unveiled the "General guidelines for assessing the design of plastic products for enhanced recyclability and regeneration" in January 2021. This regulation was introduced to address the challenges associated with recycling PPW and to tackle the issue of plastic waste pollution more comprehensively. The primary objective of this standard is to optimize recycling rates and facilitate the high-value utilization of recycled plastics. It represents China's fully autonomous design standard system for recyclable and regenerative plastic products. The forthcoming implementation of this standard will encompass certification, testing procedures, and the establishment of a safe product list and a green supply ecosystem.

*4.2. The Full-Chain Recycling Process of Plastic Packaging Waste*

The New Plastics Economy Global Commitment (NPEGC), the Clean Up Plastics Initiative (CUPI), and the European Union Global Commitment on Plastics (EUGCP) have been implemented since the launch of the EU Plastic Strategy. China and Southeast Asian countries have announced plastic bans in response to the EU Plastic Strategy's launch in January 2018. Governments and businesses are implementing measures to reduce waste and achieve a circular economy and sustainable social development throughout the product life cycle. Discarded plastics can be recycled into recycled plastics, thereby increasing the utilization of recycled plastics. Other environmentally friendly materials have been incorporated into the environmental commitments of brand companies in the automotive, electronics, electrical appliance, and packaging industries.

Plastic life cycle management [121] is used to track the entire process from the initial raw material development, product design, manufacturing, and consumption application to the final recycling and to assess the potential carbon emission impacts of all its inputs and outputs throughout its life cycle [115,122]. In addition, based on application and disposal methods, the results of plastic life cycle management direct synthesis and processing, improve processes, and enhance management in order to recycle plastics and reduce plastic pollution. Employing efficient methods in plastic's life cycle management and developing integrated technologies for the sustainable use of resources can enhance the effectiveness of plastics and mitigate their carbon emission impacts [123].

It is crucial to underscore that, within the comprehensive waste plastic recycling process, priority must be accorded to the front-end design of plastics, preceding all other stages encompassing the production, utilization, and recycling of plastic products. Beginning with the perspective of ease of recyclability and renewability in plastic products, the evaluation and guidance of plastic product design are centered on the interrelation between product design and recycling performance. Drawing on existing green packaging products in the market, one Chinese study explored the essence of green ecological design. It analyzed various methods of green design, focusing on the visual representation of product packaging, environmentally friendly materials, and structural design [124]. This strategic approach aims to surmount the obstacles hindering the enhancement of PPW recycling rates in China.

Considering the current development situation in China, the primary focus should be placed on advancing eco-friendly physical recycling technologies, enhancing chemical recycling and processing techniques for single-material waste plastics, and addressing the challenges associated with recycling and processing mixed PPW [125]. Additionally, measures should be taken to incinerate and manage hazardous PPW and plastic medical waste. [126].

In the context of China's commitment to achieving its dual-carbon goals, it is crucial to emphasize that the prevention of plastic pollution must adhere to the fundamental principles of resource conservation and environmental protection. This involves evaluating and analyzing the entire life cycle of plastic products, including their design, processing, application, and waste recycling stages. Through the development of original designs for plastic packaging products, manufacturing technology substitution, high-value utilization, and end disposal, as well as other measures, the entire PPW recycling chain can be completed.

## 5. Conclusions and Prospects

The excessive use of plastic packaging products has generated vast quantities of PPW. It is essential to strengthen the recycling and utilization of PPW and develop an economy based on plastic recycling to address the potential risks of PPW pollution. Regarding PPW management, some developed nations have provided valuable experience and mature strategies. China has also made remarkable progress in managing and recycling PPW after decades of effort. However, specific issues continue to hinder the efficiency of the PPW recycling system. Based on the principles that guide PPW management policies, we propose some management suggestions:

(1) Promote the implementation of an extended producer responsibility system in China. For instance, food delivery platforms should bear more responsibility for PPW reduction.

(2) The Chinese government should regard single-use plastic packaging as a pollutant, monitor and control the total amount from the source, and regulate the total amount at the source. Manufacturers should be prohibited from entering the market if they fail to provide production data.

(3) Measurements and statistics should be accurate and align with scientific methods and available data. This will facilitate public monitoring and the enhancement of policy decrees.

(4) The Chinese government should enhance public education on PPW recycling, particularly targeting school students. The education should focus on the necessity of recycling PPW and how to collect it.

Globally, many countries, including the United States, the UK, Canada, and China, have mainly focused on the recycling of PPW using the life cycle and circular economy concepts [127], but green designs for plastic packaging are also gradually being emphasized [118]. Therefore, the recycling process of PPW in the future will definitely be a full-chain recycling process based on green design. Furthermore, the full-chain recycling process proposed in this paper integrates green design, manufacturing, use, and recycling strategies that are very economical, environmentally friendly, and have good industrial application prospects. In the future, researchers should prioritize the green design of plastic packaging owing to its great development potential. At the same time, they should also focus their efforts on the progressive scaling up of laboratory techniques for practical applications in industry. Researchers need to continue their efforts in researching and developing green, efficient, low-carbon, and cost-effective recycling technologies for PPW. Lastly, it is imperative to intensify research on alternatives to plastic packaging to find a fundamental solution to PPW pollution.

**Author Contributions:** Conceptualization, C.Y. and G.L.; methodology, C.Y., D.J. and X.H.; validation, D.J. and W.H.; formal analysis and resources, C.Y., D.J. and W.H.; writing—original draft preparation, C.Y.; writing—review and editing, C.Y. and X.H.; supervision, W.H. and G.L. All authors have read and agreed to the published version of the manuscript.

**Funding:** This research was funded by the Shanghai Science and Technology Innovation Action Plan (22dz1209000).

**Acknowledgments:** We thank the Shanghai Environmental Sanitary Engineering Design Institute Co., Ltd., the leaders of relevant governments and enterprises, and university experts for their support and guidance.

**Conflicts of Interest:** The authors declare no conflict of interest.

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
