# Peer review of "An Overview of Management Status and Recycling Strategies for Plastic Packaging Waste in China"

_recycling, doi:10.3390/recycling8060090_

Round 1

Reviewer 1 Report

Comments and Suggestions for Authors

The paper "Management status and recycling strategies of plastic packag- 2 ing waste in China: A mini review" is of interest for the reader of the journal Recycling. However, before being considered for publication, significant changes must be made. The following comments would help to improve the authors:

Language is too colloquial

Title should be improved: "mini"

Language should be more precise: " superior performance", "Plastic is a novel material"

Each statement should be backed up by references, specially in a review paper

Figure 1 has no references

Section 2 has too many missing references, specially taking into account this a review paper

Section 3 must be improved Figure 3 is not coherent with the main texy e.g.:  primary and secondary recycling

Energy recovery should not be considered as recycling

Carbon footprint is not the same as environmental impact. section 3.5 should be more precise

Is green design a recycling strategy? Or a way to improve recycling treatments?

Figure 4 does not add value to the paper

The aspect of avoiding plastic packaging in favour of, for example, cardboard, should be analyzed in the paper

Comments on the Quality of English Language

Too colloquial language style, that should be improved

Reviewer 2 Report

Comments and Suggestions for Authors

The topic of recycling plastic products is generally of highly interesting and has been extensively covered in the literature. The manuscript describes a review of the literature on the management status and recycling strategies of plastic packing waste in China. The manuscript is a mini-review.

The authors are invited to address the two main questions about the paper: it is a geographically focused review (a mini-review of China's reality) and needs to be more critical concerning the efficiency of policies and their comparison with policies adopted in other countries.

I strongly recommend that the authors include more quantitative results of the policies implementation to improve the scientific interest of the work. The manuscript is a good technical report, but it must also be a good scientific one.

Figures and tables should always be placed close to their reference in the text.

Reviewer 3 Report

Comments and Suggestions for Authors

The manuscript by Yu et al. ‘Management status and recycling strategies of plastic packaging waste in China: A Review’ describes the plastic waste management situation in China in part. This first part gives interesting information, which is well worth reporting. However, in the perspective sections starting at 3.3 the actual situation is not described anymore, but a more generic review about improvements and assessment are reported, that are not specific to the situation in China. Even if the information is not available, it would be critical, that the authors address, what could be adopted by China, and maybe also when. Adding this specifics is necessary to achieve the goal of the paper at least according to the title and the abstract. Therefore, I suggest a major revision to specify the chapter 3.3 and following to the case of China. Some minor comments can be find below.

Regarding language, I see no need for improvements.

L75: I think the authors could combine the sentence with the one before.

L82-83: This statement is not conclusive, how can there be a ‘calling for a gradual phase-out of non-degradable plastics’ and a at the same time ‘improvement of the production, consumption, recycling, and disposal mechanisms for plastic products’. This statements sounds as if China intends to ban all plastics that are non-degradable. Is that the case. Furthermore do the authors mean biodegrdable or degradable, which would include chemical and physical processes that affect any type of plastic.

L210: spelling ‘according’

L301-303. The reference is missing and the message is not conclusive. Which periode do you consider for the epidemic, different countries had different definitions when it ended. Also it it is not clear, how the numbers compare.

L283: I’m missing some numbers on China in comparison in the section of Energy recycling.

L423-426: The term ‘we’ seems out of context here, bis check the sentence.

L408: I’m missing also in this chapter the connext to China. The paragraph appears more like a summary of measures that can be taken, but no connex to China is made here.

Basically the same applies to the conclusion section.

Comments on the Quality of English Language

Some minor changes are necessary, please look above.

Round 2

Reviewer 1 Report

Comments and Suggestions for Authors

The authors have properly replied to all my comments. My recommendation is Accept

Comments on the Quality of English Language

The authors have properly replied to all my comments. My recommendation is Accept

Reviewer 2 Report

Comments and Suggestions for Authors

The authors have performed significant changes in the manuscript.

Reviewer 3 Report

Comments and Suggestions for Authors

The authors, adressed all suggestions accordiningly and can recommend the article for publication

I would only suggest they replace in L336 the word gas with steam.